# Real-Time Trajectory Smoothing and Obstacle Avoidance: A Method Based on Virtual Force Guidance

**DOI:** 10.3390/s24123935

**Published:** 2024-06-18

**Authors:** Yongbin Su, Chenying Lin, Tundong Liu

**Affiliations:** Pen-Tung Sah Institute of Micro-Nano Science and Technology, Xiamen University, Xiamen 361104, China; suyongbin@stu.xmu.edu.cn (Y.S.); linchenying@stu.xmu.edu.cn (C.L.)

**Keywords:** trajectory planning, real-time trajectory smoothing, real-time obstacle avoidance, virtual force guidance

## Abstract

In dynamic environments, real-time trajectory planners are required to generate smooth trajectories. However, trajectory planners based on real-time sampling often produce jerky trajectories that necessitate post-processing steps for smoothing. Existing local smoothing methods may result in trajectories that collide with obstacles due to the lack of a direct connection between the smoothing process and trajectory optimization. To address this limitation, this paper proposes a novel trajectory-smoothing method that considers obstacle constraints in real time. By introducing virtual attractive forces from original trajectory points and virtual repulsive forces from obstacles, the resultant force guides the generation of smooth trajectories. This approach enables parallel execution with the trajectory-planning process and requires low computational overhead. Experimental validation in different scenarios demonstrates that the proposed method not only achieves real-time trajectory smoothing but also effectively avoids obstacles.

## 1. Introduction

Trajectory planning entails determining a path between two points within a specified region while avoiding collisions with environmental objects [1]. It is pivotal for various unmanned platforms, including industrial robots, mobile robots, and autonomous drones. The smoothness of trajectories is a critical performance metric in trajectory planning, as jerky trajectories can lead to frequent transitions between “stop”, “rotate”, and “restart” motion states, resulting in slip and over-responsive behavior during high-speed movements [2,3]. In dynamic environments, unpredictable changes occur frequently, necessitating online adjustments or recalibrations of paths to ensure safe navigation around newly detected nearby objects [4]. Hence, there is a critical need for efficient online trajectory-smoothing and obstacle-avoidance methods to meet the post-processing requirements of real-time planning based on sampling.

To address trajectory planning in dynamic environments, unmanned platforms need to identify potential paths based on obstacle information at each time step and determine the next foothold until reaching the target position [5]. While real-time trajectory planning allows adaptive responses to sudden environmental changes, the lack of smoothness makes them inefficient and prone to system instability [6]. Smooth trajectories are desirable in trajectory planning for practical unmanned platforms as they appear more predictable and satisfactory to humans [7]. Existing trajectory-smoothing methods can be categorized into global and local smoothing approaches [8]. Global smoothing fits all trajectory points using a parameterized curve (e.g., NURBS [9], polynomial splines [10], or B-splines [8]) after obtaining the complete trajectory, which is impractical for real-time trajectory-planning scenarios. Local smoothing methods aim to replace sharp corners within two adjacent line segments with smooth blending curves. Various local smoothing algorithms have been proposed, incorporating different curve representations to ensure smooth transitions at trajectory corners [11,12,13,14,15,16]. However, these methods may produce paths that collide with obstacles (as indicated in the situation shown in Figure 1) since the smoothing process lacks a direct connection with trajectory optimization [15].

To address the real-time smoothing and obstacle-avoidance problem, many researchers have studied real-time trajectory smoothers based on shortcut techniques. The idea of shortcutting a smoothed path was initially proposed by Geraerts and Overmars [17]. Hauser et al. [18] extended this approach by introducing parabolic trajectory representations to consider velocity and acceleration bounds. Subsequently, Ran et al. [19] further expanded this parabolic smoothing algorithm to incorporate jerk constraints. Furthermore, Pan et al. [20] introduced spline-based trajectory representation and its associated shortcut algorithm. Although all of these methods successfully compute smooth paths for piecewise linear trajectories, their computation speed is slow due to numerous collision checks. To optimize computation speed, Shohei et al. [6] employed neural networks to estimate gaps and collisions between a robot and voxels in a parallel manner, combined with shortcut techniques to construct a real-time trajectory smoother, achieving a cycle time of 300 ms. However, this approach requires high hardware requirements and extensive dataset collection for training the network model, limiting its practicality.

Here, we propose a novel trajectory-smoothing framework to provide real-time trajectory smoothing for trajectory planners based on real-time sampling, achieving obstacle avoidance and trajectory smoothing in dynamic environments. This method constructs a “follower” to track the trajectory points generated by the trajectory planner, where points ahead of the “follower” exert attractive forces, pulling it to follow the original trajectory, while points behind the “follower” exert repulsive forces, pushing it forward. When obstacles appear, a repulsive force is applied to the “follower” to encourage it to move away from the obstacles. The resultant force from all forces determines the forward direction of the “follower”. By appropriately setting the follower’s velocity and introducing a lag effect at corners, trajectory smoothing is achieved. The position of the “follower” is recorded at each time step, constituting the final smoothed trajectory.

## 2. Proposed Method

### 2.1. The Overall Framework of the Proposed Method

The overall framework of the method proposed in this paper is illustrated in Figure 2. It explains the effectiveness, principles, and processes of this method. The performance of the proposed real-time trajectory smoothing method based on virtual force guidance is shown in the upper left part of Figure 2. The trajectory planner in the figure adopts a real-time sampling-based trajectory-planning method, which calculates the position of the next point at each subsequent time step until reaching the goal point while avoiding obstacles. The trajectory smoother, employing the method proposed in this paper, operates concurrently with the trajectory planner. Whenever the planner generates a new sampling point, the smoother incorporates it and calculates the next position using the smoothing algorithm.

The smoother first defines a “follower” that follows the trajectory points generated by the trajectory planner from time $t_{0}$ according to certain rules. Specifically, as shown in the upper right part of Figure 2, the point generated by the planner at a certain moment and the points generated in the previous three time steps (labeled *A*, *B*, *C*, and *D*, respectively), along with the obstacles, jointly influence the output of the smoother. Points *A* and *B* exert attractive forces on the “follower”, while the other points exert repulsive forces. The resultant force guides the “follower” to the position at the subsequent time step, continuing until reaching the goal point. The overall process of this method is shown in the lower part of Figure 2.

### 2.2. Analysis of Obstacle-Free Scenarios

A concept known as the reference point is introduced into the original trajectory (generated by the trajectory planner), akin to the role of point *B* in Figure 2; this serves as the primary attractor for the “follower”. As shown in Figure 3, assuming that the red point represents the “follower” (the current position of the unmanned platform, i.e., the goal point of the smoothed trajectory), denoted as *P*, and the blue point represents the reference point of the original trajectory, denoted as $x_{k}$, with its subsequent point being denoted as $x_{k+1}$ and its preceding points being denoted as $x_{k-1}$ and $x_{k-2}$. The forces that they exert on point *P* are denoted as follows.
(1)$${\vec{F}}_{k}=k_{at1}\cdot \left(x_{k}-P\right)$$
(2)$${\vec{F}}_{k+1}=k_{at2}\cdot \left(x_{k+1}-P\right)$$
(3)$${\vec{F}}_{k-1}=k_{re1}\cdot \left(x_{k-1}-P\right)$$
(4)$${\vec{F}}_{k-2}=k_{re2}\cdot \left(x_{k-2}-P\right)$$
where $k_{at1}$, $k_{at2}$, $k_{re1}$, and $k_{re2}$ represent the coefficients of attractive and repulsive forces; the resultant force $F_{ori}$ experienced by the “follower” can be calculated using the following formula.
(5)$$\vec{F_{ori}}=\vec{F_{k}}+\vec{F_{k+1}}+\vec{F_{k-1}}+\vec{F_{k-2}}$$

The direction of vector $\vec{F_{ori}}$ serves as the direction of motion for *P*, and its magnitude determines the motion step length. *P* will move forward to reach the position of the red point in Figure 4a. If, at this point, the reference point remains unchanged, the process continues with force analysis controlling the forward movement of *P*, as described above. If the reference point updates to the next point (i.e., changing from the original $x_{k}$ to $x_{k+1}$), then *P* undergoes force analysis and motion under the influence of the new reference point, as shown in Figure 4b. The specific strategy for updating the reference point will be discussed in Section 2.4.

### 2.3. Analysis in the Presence of Obstacles

For irregularly shaped obstacles, the current mainstream approach is to replace them with a circumcircle [21], as circular obstacle boundaries are more conducive to generating smooth trajectories. However, if only a circumcircle of obstacles is constructed to replace the original shape, it will encroach on a significant amount of free space, resulting in overly lengthy generated trajectories. As shown in Figure 5a, the circumcircles of two irregularly shaped obstacles block the gap between them, forcing the planned trajectory to detour around them (solid lines in the figure), rather than passing through the gap (dashed lines in the figure), which is undesirable.

To address this issue, we adopt a more efficient approach by rounding the corners of irregular obstacles with circles of radius *r*, as shown in Figure 5b. This not only makes the obstacle surface smoother, facilitating subsequent algorithm execution, but also minimizes the space occupied, thus preserving the optimal solution as much as possible. When the position *P* of the “follower” is within the influence range of obstacles, it is necessary to introduce the effect of obstacle repulsion to encourage the points generated by the trajectory smoother to move away from obstacles. The black arrows in the figure represent the direction of the repulsive force exerted by the obstacles on the surroundings, which are perpendicular to the contours of the obstacles. When the “follower” approaches the range of action of the obstacles, it will be affected by the repulsive force of the obstacles. The point where the repulsive force is generated is the point on the contour of the obstacle that is closest to it. If there is more than one such point, then the repulsive force is the resultant force of all points acting on it. As shown by the blue dashed circle in the figure, suppose that the “follower” moves to the position where the red dot is located. There are two points on the contour of the obstacle closest to it, so it will be affected by two forces, $F_{1}$ and $F_{2}$. The final repulsive force exerted by the obstacle on it is the resultant force of these two forces, denoted as $F_{obs}$.

For the sake of simplicity in modeling and description, in the subsequent content, we still assume that all obstacles are circular. Discussing the general case, when the planned trajectory passes through obstacles, as shown in Figure 6a, in addition to the forces exerted on point *P* by the four points of the original trajectory, the obstacles also exert forces on it. The resultant force of these components guides the extension direction of the smooth trajectory.

Similarly to the artificial potential field (APF) method, obstacles typically have an influence range ρ, and only objects within this range are affected by the repulsive force of the obstacles. However, there is a scenario where the “follower” enters the influence range of the obstacles, but its direction of motion has never pointed toward the obstacles and just passes by. If such instances are influenced by the obstacles, it can lead to unnecessary deviations in the smooth trajectory. Therefore, this study determines whether the obstacles will affect the “follower” by examining the number of intersection points between the direction of motion of the “follower” and the contour of the obstacles.

Suppose that the influence radius of the obstacle is ρ, as shown in Figure 6b. At time *k*, the position of the “follower” is $y_{k}$, and its position at the previous time is $y_{k-1}$. Connecting these two points into a straight line, it is assumed that they intersect with the obstacle at points *M* and *N* and intersect with the boundary of the influence range at point *O*. If points *M* or *N* exist, then when analyzing the force acting on $y_{k}$, the repulsive force will be considered. We define the distance from $y_{k}$ to the nearest intersection point *O* as $d_{1}$ and the distance from *O* to *M* as $d_{2}$. Then, the repulsive force acting on $y_{k}$ is defined as
(6)$$|\vec{F_{obs}}|=d_{1}/d_{2}*F_{max}$$
where $F_{max}$ represents the maximum value of the repulsive force, that is, the value when $y_{k}$ comes into contact with the surface of the obstacle. It is not fixed but has a certain relationship with $F_{ori}$, $F_{max}=k*F_{ori}$, to prevent sudden changes in the final resultant force when approaching the obstacle. The direction of $\vec{F_{obs}}$ is from the center of the obstacle to $y_{k}$. If there are no intersection points, then the repulsive force at this time will be zero.

### 2.4. Reference Point Update Strategy

In the absence of obstacles, relying solely on the virtual force guidance provided by the original trajectory, the “follower” may closely track the original path, potentially overlapping with it, thus failing to achieve the desired smoothing effect. To address this issue, we introduce a maximum tracking error parameter disErr to ensure that the original trajectory slightly leads the smoothed trajectory. Consequently, when the original trajectory encounters corners, the smoothed trajectory, due to its inherent lag, can smoothly transition around them. If the distance between the reference point of the original trajectory and the “follower” exceeds a certain threshold, the update frequency of the reference point is reduced by a factor of $v_{0}$ until their distance meets the tracking error requirement. This approach not only ensures smooth transitions around corners but also prevents excessive distortion of the smoothed trajectory.

When approaching obstacles, the update of the reference point needs to be redesigned. If the sampling points output by the trajectory planner are too close to obstacles and are obstructed by circles covering the corner points, the original trajectory may penetrate through the obstacles. In such cases, the smoothed trajectory needs to navigate around the obstacles. This may result in a longer path compared to the original trajectory, and using the update strategy from the obstacle-free scenario may lead to oscillations in the output points of the trajectory smoother. Our strategy is to adjust the update frequency of the reference point based on the distance between the reference point and the center of the obstacle, categorized into three levels $\left\{v_{1},v_{2},v_{3}\right\}$, as illustrated in Figure 7. Specifically, the update frequency is accelerated when entering the influence range of obstacles, reduced when close to the center of the obstacles, and restored to the baseline update frequency when moving away from obstacles.

In addition to setting the maximum tracking error and the update frequency of the reference point, it is necessary to define a maximum step size parameter stepMax to adjust the feed rate of the “follower” to prevent abrupt changes in trajectory caused by excessively large step sizes. If the calculated feed rate exceeds this limit, the step size in each direction is proportionally reduced. Furthermore, since the forward direction and step size of the “follower” are guided by the resultant force, situations where the resultant force equals zero may occur, which are known as local optima in the APF method. To avoid this, a small force $\vec{F_{1}}$ perpendicular to $\vec{F_{ori}}$ is introduced into the force calculation. Algorithm 1 outlines this process.
**Algorithm 1:** Real-Time Smoothing Algorithm
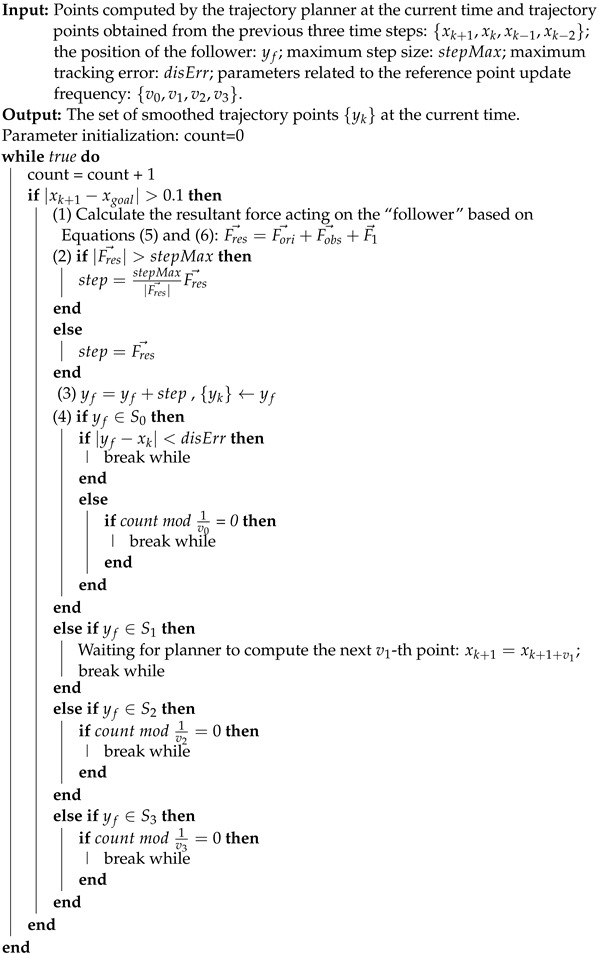


## 3. Experimental Results

### 3.1. Experimental Setup and Evaluation Metrics

We conducted simulation experiments in a two-dimensional plane. The starting point is represented by blue dots, the destination point is represented by red dots, and obstacles are represented by black-filled circles with shaded regions indicating their influence range. Since we focus solely on the smoothing process of known trajectories rather than trajectory planning, we preset the trajectory path and simulate the real-time planning process, where yellow dots represent points generated at each time step. Our trajectory smoother processes these yellow points in real time, smoothing them at corners; these are represented by blue dots in the experiment. We only present the final processed trajectory, while the specific process can be observed by running our open-source code (https://github.com/syb-xmu/Real-time-Smoothing.git).

In the absence of obstacles, local smoothing methods such as Bezier curves effectively smooth corners. Therefore, in this scenario, we focus on analyzing the influence of our algorithm’s parameters on the experimental results. In the presence of obstacles, we compare our method with RRT–Bezier-curve-based and APF-based methods. However, since these methods require the destination to be known beforehand, we conduct obstacle-avoidance processing only after the trajectory planner passes through obstacles. We define three quantitative evaluation metrics to assess our algorithm. The curvature measures trajectory smoothness and is calculated as described in [22]. The Fréchet distance [23] evaluates the similarity between the original and smoothed trajectories. The collision probability assesses the algorithm’s obstacle-avoidance performance and is calculated as the ratio of collision instances to total experimental trials.

### 3.2. Case Study 1: No-Obstacle Scenario

In this section, we explore the impacts of two critical parameters, namely, the maximum step size and the maximum tracking error, on the algorithm’s performance. We assume that the trajectory planner generates a trajectory resembling a sawtooth wave, and our algorithm performs real-time smoothing around corners. First, we investigate the influence of different maximum tracking errors, setting their values to 0.5, 1.0, and 2.0, while keeping other parameters constant: $k_{at1}=1$, $k_{at2}=0.5$, $k_{re1}=k_{re2}=0.01$, k=1.1, $v_{0}=\frac{1}{3}$, $v_{1}=2$, $v_{2}=\frac{1}{6}$, $v_{3}=\frac{1}{2}$, and stepMax=0.1. The obtained results and corresponding curvature are shown in Figure 8.

Since the number of points in the smoothed trajectory varies for different parameters, the relative positions of the smoothed trajectory and the original trajectory in the curvature plot also differ. From the results, it can be observed that a larger maximum tracking error leads to smoother trajectories but also introduces greater distortion. Specific quantitative metrics are shown in Table 1, indicating an inverse relationship between trajectory smoothness and distortion. Hence, suitable parameters can be chosen to balance these factors according to practical requirements.

In our algorithm, the maximum step size is also a crucial parameter affecting trajectory smoothness. For the same corner, if the maximum step size is too large, smooth transitions cannot be achieved; if it is too small, the trajectory may not keep up, leading to distortion. We further investigate the influence of different maximum step sizes, setting their values to 0.12, 0.15, and 0.18 while setting disErr=1, with the other parameters remaining unchanged. The obtained results and corresponding curvature are depicted in Figure 9.

From the results, it can be observed that larger maximum step sizes result in more pronounced corners in the trajectory, indicating poorer smoothness. Specific quantitative metrics are shown in Table 2, where an increase in the maximum step size leads to a closer match between the extension speed of the smoothed trajectory and the original trajectory, without any lag effect. Consequently, although the trajectories exhibit high similarity, the smoothness is compromised.

### 3.3. Case Study 2: Single-Obstacle Straight Trajectory Scenario

This experiment investigates the performance of our method in a simple scenario where the trajectory passes through an obstacle in a straight line. We compare our method with traditional RRT algorithms, RRT algorithms with Bezier curves, and traditional APF methods to validate the superior obstacle avoidance and smoothing performance of our approach. Traditional RRT algorithms, due to their randomness, yield different results each time and may generate trajectories with peculiar shapes, as shown in Figure 10a. Incorporating Bezier curves can achieve smoothing effects, but collisions with obstacles may occur, as illustrated in Figure 10b, because the post-smoothing process is not directly connected to the trajectory-planning process and does not consider obstacle constraints. We ran the RRT algorithm 20 times and selected the best results for comparison with our method, as depicted in Figure 10c. Additionally, we calculated the collision probability by counting the number of collisions that occurred after smoothing with Bezier curves.

The trajectory generated by the traditional APF method tends to deviate significantly from obstacles due to the lack of constraints, resulting in a large deviation from the ideal trajectory. To ensure fairness, we also use Equation (Equation 6) to replace the traditional APF method’s obstacle repulsion calculation. Additionally, to adapt the traditional APF method to our scenario, we set the entry point into the obstacle’s influence area as the starting point and designate the exit point as the goal. Between these two points, we employ the APF method for the transition. The trajectories obtained with traditional APF methods and our algorithm, as shown in Figure 11a,b, reveal that the trajectories generated by traditional APF methods exhibit a directional discontinuity when approaching the lowest point of the obstacle, while our method smoothly circumvents the obstacle. The curvature plots obtained from these four methods, as shown in Figure 11c, demonstrate that our method has the lowest maximum curvature.

Comparing the results in Figure 10c with those in Figure 11a,b, we observe that among these four methods, the trajectories obtained with the RRT-based method can easily avoid obstacles, but they appear stiff, and collisions may occur after smoothing with Bezier curves. The traditional APF method ensures collision-free trajectories but exhibits directional discontinuities at specific locations, resulting in non-smooth trajectories. Table 3 presents a quantitative comparison of the results of different methods. In terms of trajectory similarity, both the traditional APF method and our method demonstrate superior performance. However, our method has a lower maximum curvature, indicating better smoothness. Additionally, the traditional RRT- and Bezier-curve-based methods experienced collisions in 6 out of the 20 experiments, indicating a significant risk despite their superior smoothness. In summary, our method outperforms others in terms of trajectory similarity and smoothness, and most importantly, it achieves real-time smoothing effects, while other methods require separate obstacle avoidance and smoothing processes after the trajectory planner completely bypasses the obstacles.

### 3.4. Case Study 3: Single-Obstacle Corner Trajectory Scenario

To minimize the collision space caused by obstacles and to eliminate some of the sharp corners of the original obstacles, we use circular obstacles to cover the corners of the original obstacles (as described in Section 2.3). However, this inevitably obstructs the trajectories planned by the trajectory planner. Therefore, in this experiment, we explore the obstacle avoidance and smoothing performance of our algorithm with different corner angles. As the previous section’s experiment showed, the trajectory smoothness of the traditional RRT algorithm is poor, and adding Bezier curves may lead to collisions with a relatively high probability. Therefore, in this section, we only compare our method with the APF method.

Corner angles of 45 degrees, 90 degrees, and 135 degrees are set, and the results are shown in Figure 12. It can be seen that the trajectories generated by our method smoothly navigate around the obstacles, and the trajectory directions are reasonable. However, the traditional APF method cannot consider a trajectory inside the obstacle, so it directly selects the original destination point (i.e., the point where the original trajectory exits the obstacle) as the attractor point, resulting in an unreasonable trajectory. In contrast, our method dynamically follows the virtual target points of the original trajectory, so it can reproduce the trajectory’s path as much as possible, making the smoothed trajectory more reasonable and of higher quality.

### 3.5. Case Study 4: Multi-Obstacle Complex Trajectory Scenario

Our method also applies to situations where unexpected obstacles suddenly appear in the original trajectory, which is common in the field of learning from demonstrations. In this experiment, we assume that the original trajectory follows a sinusoidal function shape, and at some point, two obstacles suddenly appear. Using our method, the trajectory can smoothly navigate around these obstacles. Figure 13 illustrates two scenarios with obstacles at different positions. In Figure 13a, the obstacle on the left does not obstruct the original trajectory directly, but the original trajectory enters its influence range. However, since we only consider repulsive forces when the direction of motion intersects with the corner of the obstacle, as described in Section 2.3, entering the influence area of the obstacle does not change the trajectory’s direction.

## 4. Discussion and Conclusions

This paper proposes a real-time trajectory-smoothing framework based on virtual force guidance that can be executed in parallel with real-time trajectory-planning algorithms. By constructing a virtual force model between the original trajectory and obstacles, it guides the real-time generation of smooth trajectories. In the absence of obstacles, the proposed method can smoothly navigate the corners of the original trajectory. When obstacles appear, comparisons with RRT and APF algorithms reveal that our method not only smoothly navigates obstacles but also maintains the direction of the original trajectory well, successfully avoiding obstacles and, thereby, validating the effectiveness and superiority of our method.

Our method is not only applicable to real-time trajectory smoothing but also to real-time smoothing of sampled signals to remove signal spikes and high-frequency noise. Additionally, our method is suitable for obstacle avoidance in pre-planned trajectories, especially in the field of demonstration learning. If pre-planned trajectories are established and obstacles suddenly appear during execution, our method can effectively avoid them. In the absence of obstacles, simply reducing the maximum tracking error can essentially replicate the original trajectory. Furthermore, by appropriately setting the maximum step size, the acceleration and deceleration processes can be seamlessly integrated into the proposed method without the need for subsequent interpolation processing, making it highly suitable for practical applications in unmanned platforms.

## Figures and Tables

**Figure 1 sensors-24-03935-f001:**
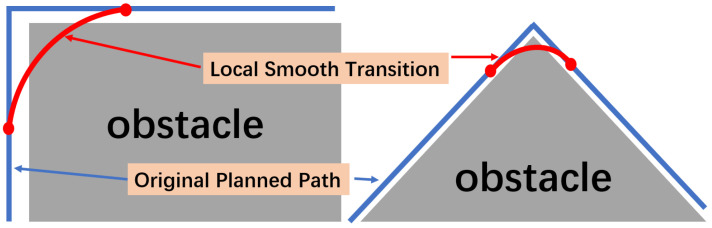
The collision situations between the trajectories generated by the local smoothing algorithm and obstacles.

**Figure 2 sensors-24-03935-f002:**
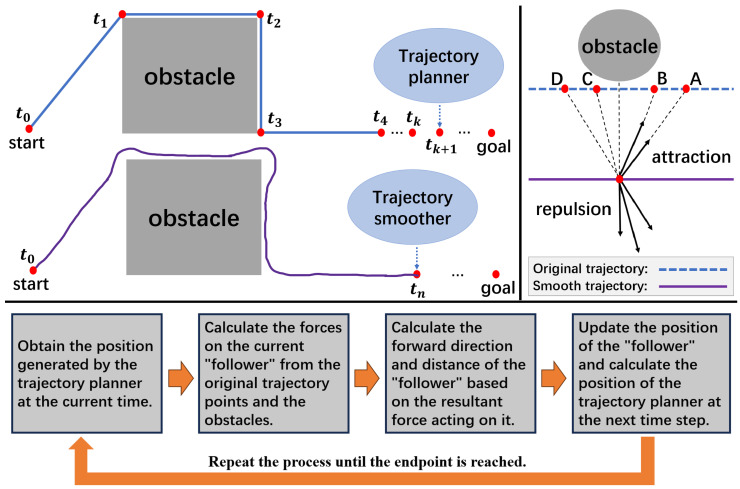
The overall framework of the proposed method.

**Figure 3 sensors-24-03935-f003:**
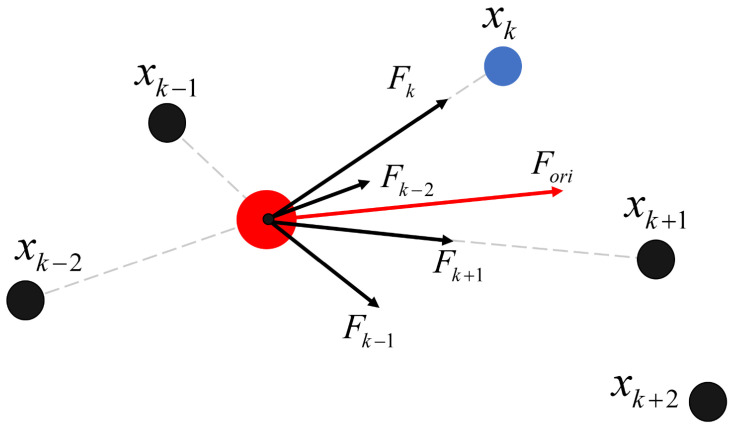
Analytical diagram of the virtual forces exerted on the “follower” by points in the original trajectory.

**Figure 4 sensors-24-03935-f004:**
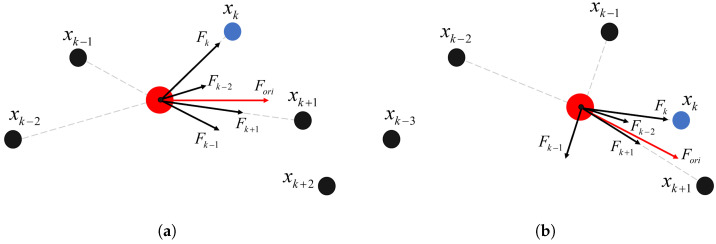
Analysis of the forces acting on the “follower” and its motion at different reference points. Panel (**a**) depicts the force analysis diagram when the reference point remains unchanged. Panel (**b**) illustrates the force analysis diagram when the reference point is updated to the next trajectory point.

**Figure 5 sensors-24-03935-f005:**
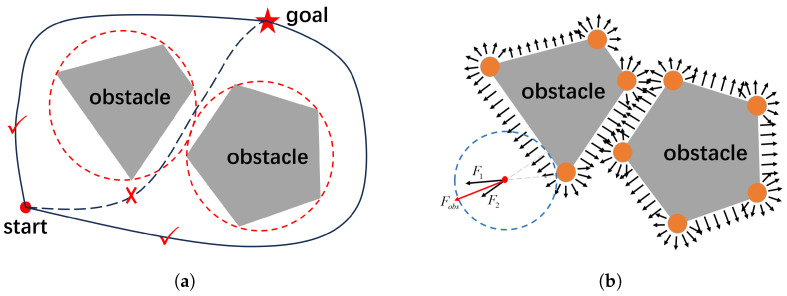
Different approaches to handling irregular obstacle corners. Panel (**a**) depicts covering the original obstacle with a circumscribed circle, allowing the trajectory to pass only around the obstacle edges and making it unable to penetrate through the center. Panel (**b**) illustrates placing a circular obstacle at each corner of the obstacle and analyzing the force acting on objects within the obstacle’s influence range.

**Figure 6 sensors-24-03935-f006:**
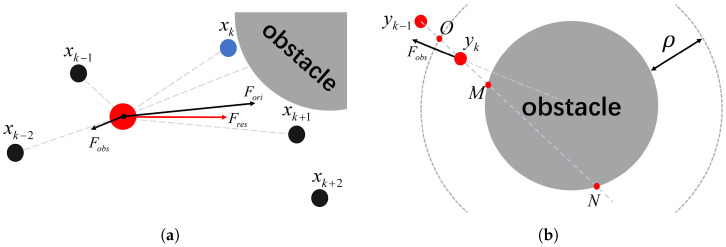
Force analysis in the presence of obstacles. Panel (**a**) depicts the overall force acting on the “follower” when obstacles are introduced. Panel (**b**) separately analyzes the calculation of the magnitude and direction of the force exerted by obstacles on the follower.

**Figure 7 sensors-24-03935-f007:**
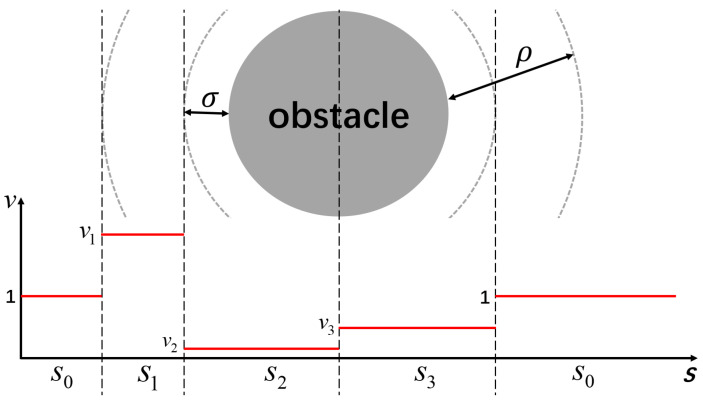
The update frequency of the reference points from the original trajectory within the range of influence of obstacles.

**Figure 8 sensors-24-03935-f008:**
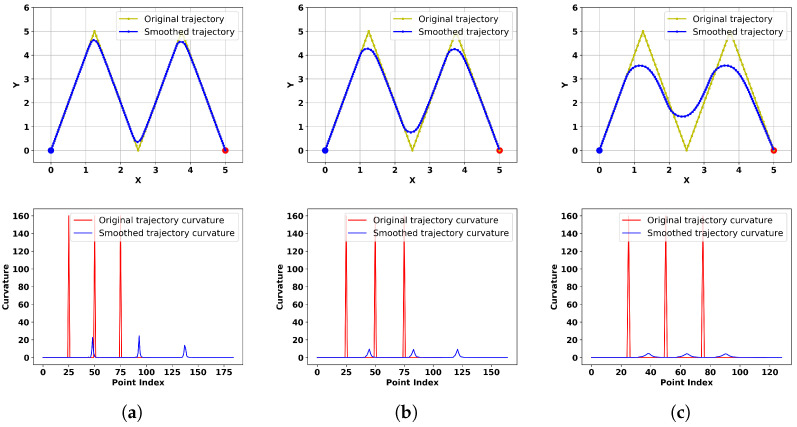
The smoothed trajectory directions and corresponding curvature values with different maximum tracking errors (**a**–**c**).

**Figure 9 sensors-24-03935-f009:**
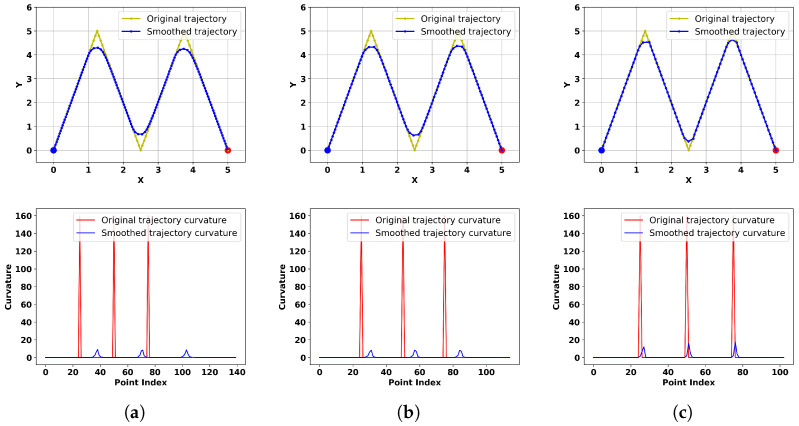
The smoothed trajectory directions and corresponding curvature values with different maximum step sizes (**a**–**c**).

**Figure 10 sensors-24-03935-f010:**
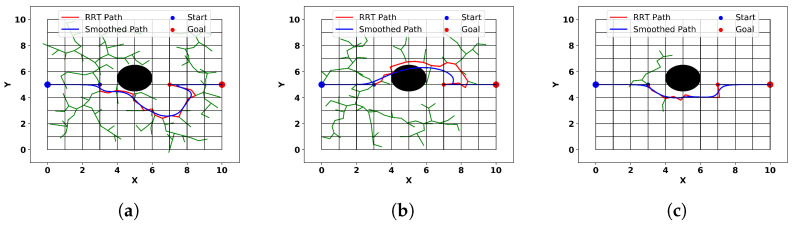
The trajectories obtained from the smoothing methods based on the RRT algorithm and Bezier curves.

**Figure 11 sensors-24-03935-f011:**
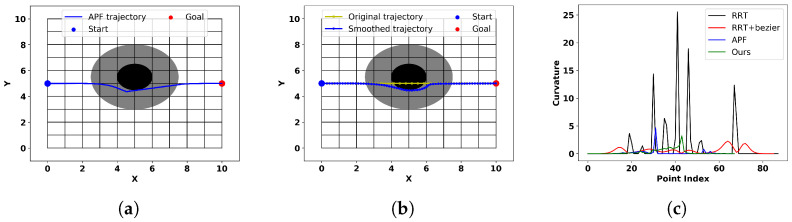
(**a**) Trajectory obtained from artificial potential field (APF). (**b**) Trajectory obtained from the proposed method. (**c**) Curvatures of trajectories obtained from different methods.

**Figure 12 sensors-24-03935-f012:**
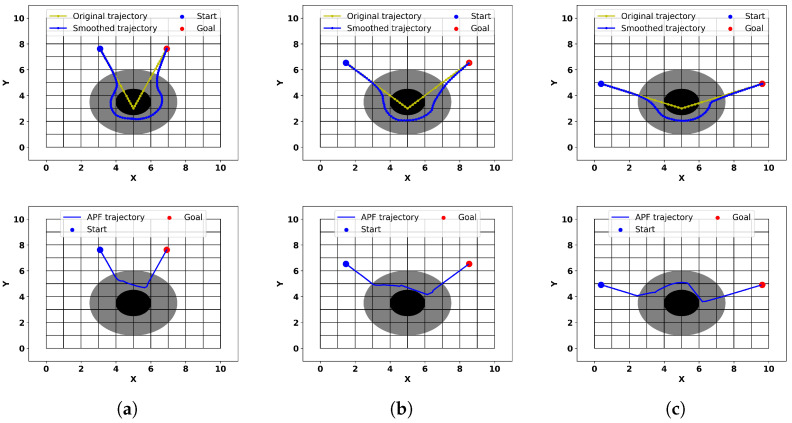
Smooth trajectories obtained with the proposed method and the APF method at different angles (**a**–**c**).

**Figure 13 sensors-24-03935-f013:**
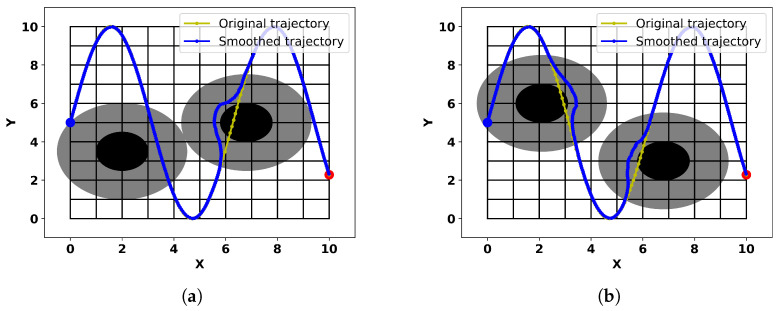
Smooth obstacle-avoidance trajectories generated by the proposed method at different obstacle positions (**a**,**b**).

**Table 1 sensors-24-03935-t001:** The trajectory similarity and smoothness with different maximum tracking errors.

Maximum Tracking Error	Fréchet Distance	Max Curvature
0.5	0.45	24.45
1.0	0.76	9.49
2.0	1.45	4.47

**Table 2 sensors-24-03935-t002:** The trajectory similarity and smoothness with different maximum step sizes.

Maximum Step Size	Fréchet Distance	Max Curvature
0.12	0.75	9.06
0.15	0.68	8.33
0.18	0.49	17.49

**Table 3 sensors-24-03935-t003:** Trajectory similarity, smoothness, and collision probability with different methods.

Method	Fréchet Distance	Max Curvature	Collision Probability
RRT	1.19	25.56	0%
RRT + Bezier	1.03	2.25	30%
APF	0.64	4.63	0%
Ours	0.56	3.22	0%

## Data Availability

Data are contained within the article. We have open-sourced our code on GitHub at https://github.com/syb-xmu/Real-time-Smoothing.git.

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
