# Peer review of "Real-Time Trajectory Smoothing and Obstacle Avoidance: A Method Based on Virtual Force Guidance"

_sensors, 2024, doi:10.3390/s24123935_

Round 1
Reviewer 1 Report
Comments and Suggestions for Authors
This paper proposes a novel trajectory smoothing method that considers obstacle constraints in real-time. The topic in this paper is interesting, and it has some potential applications. This paper could be considered for publication but subject to some necessary revisions:
(1) What is the difference between the method in this paper and the artificial potential field method?
(2) Normally, the obstacle avoidance trajectory of the artificial potential field method is relatively smooth. Why is the trajectory in Figure 11a not like this?
(3) Are the obstacles in the simulation experiment static? How does the method proposed in this paper perform in dynamic obstacle avoidance environments?
(4) Although some methods have been compared in the current paper, they are inconsistent with the main contribution points of this paper. The author should compare it with existing trajectory smoothing methods, rather than path planning methods.
(5) Regarding the literature survey, the reviewer recommends to add a few more papers related to this study. Such as: Lu E, Tian Z, Xu L, et al. Observer-based robust cooperative formation tracking control for multiple combine harvesters [J]. Nonlinear Dynamics, 2023, 111(16): 15109-15125.
(6) Some grammar & format errors and typos need to be modified in the revised version.
Comments on the Quality of English LanguageMinor editing of English language required
Reviewer 2 Report
Comments and Suggestions for Authors
This paper proposes a novel trajectory smoothing method that considers obstacle constraints in real-time. By introducing virtual attractive forces from original trajectory points and virtual repulsive forces from obstacles, the resultant force guides the generation of smooth trajectories. There are some comments need to be clarfied in the paper.
1. The overall framework of the method in Figure 2 is not clear and specific enough. It should describe how to deal with each step clearly.
2. How to calculate all the F in 2.2 is not clear .
3. The article proposes a method to deal with irregular obstacles, but the effectiveness of this method has not been verified. Please explain further.
4. What is the value of Fobs based on? Please further explain formula 2.
Round 2
Reviewer 2 Report
Comments and Suggestions for Authors
The author revised the paper accordingly, and it can be published as the current version.